# Brownian Noise Reduction:
# Maximizing Privacy Subject to Accuracy Constraints

**Justin Whitehouse**
Carnegie Mellon University
jwhiteho@andrew.cmu.edu

**Zhiwei Steven Wu**
Carnegie Mellon University
zstevenwu@cmu.edu

**Aaditya Ramdas**
Carnegie Mellon University
aramdas@cmu.edu

**Ryan Rogers**
LinkedIn
rrogers@linkedin.com

## Abstract

There is a disconnect between how researchers and practitioners handle privacy-utility tradeoffs. Researchers primarily operate from a *privacy first* perspective, setting strict privacy requirements and minimizing risk subject to these constraints. Practitioners often desire an *accuracy first* perspective, possibly satisfied with the greatest privacy they can get subject to obtaining sufficiently small error. Ligett et al. [2017] have introduced a "noise reduction" algorithm to address the latter perspective. The authors show that by adding correlated Laplace noise and progressively reducing it on demand, it is possible to produce a sequence of increasingly accurate estimates of a private parameter while only paying a privacy cost for the least noisy iterate released. In this work, we generalize noise reduction to the setting of Gaussian noise, introducing the *Brownian mechanism*. The Brownian mechanism works by first adding Gaussian noise of high variance corresponding to the final point of a simulated Brownian motion. Then, at the practitioner's discretion, noise is gradually decreased by tracing back along the Brownian path to an earlier time. Our mechanism is more naturally applicable to the common setting of bounded $\ell_2$-sensitivity, empirically outperforms existing work on common statistical tasks, and provides customizable control of privacy loss over the entire interaction with the practitioner. We complement our Brownian mechanism with $\mathrm{ReducedAboveThreshold}$, a generalization of the classical $\mathrm{AboveThreshold}$ algorithm that provides adaptive privacy guarantees. Overall, our results demonstrate that one can meet utility constraints while still maintaining strong levels of privacy.

## 1 Introduction

Over the past decade, differential privacy has seen industry-wide adoption as a means of protecting sensitive information [Erlingsson et al., 2014, Greenberg, 2016]. By injecting appropriate amounts of noise, differentially private algorithms allow the computation of population-level quantities of interest while guaranteeing individual-level privacy. Of the private mechanisms used in industry, those relating to private empirical risk minimization (ERM) are perhaps the most impactful, in part due to their application in machine learning tasks [Abadi et al., 2016, Song et al., 2013]. Researchers have developed many private ERM mechanisms, ranging from least squares minimzation [Sheffet, 2017, Chaudhuri et al., 2011] to subsampled gradient descent [Abadi et al., 2016, Balle and Wang, 2018, Wang et al., 2019]. Despite this vast literature, most existing results take the same broad approach: they aim to minimize error (statistical risk) subject to strict privacy guarantees. While this strict adherence to privacy constraints may be necessary in some applications, it often provides weak

36th Conference on Neural Information Processing Systems (NeurIPS 2022).

utility guarantees [Fienberg et al., 2010] and can make some learning tasks impossible [Dwork et al., 2009]. Industry applications of differential privacy may desire an *accuracy first* perspective, setting desired risk requirements for models used in production. Privacy may still be a desirable aspect of computation, but it is by no means the only goal; minimizing risk may take center stage.

The main existing approach to this accuracy-oriented perspective on privacy was given by Ligett et al. [2017]. These authors introduce a *noise reduction mechanism* for gradually releasing a private, high-dimensional parameter. By leveraging a Laplace-based Markov process [Koufogiannis et al., 2017], they construct a mechanism for which the privacy loss of releasing arbitrarily many estimates of a parameter only depends on the privacy loss of the least noisy parameter viewed. This is in contrast to results about the composition of private algorithms, in which privacy degrades according to the total number of parameters witnessed [Dwork et al., 2010, Kairouz et al., 2015, Murtagh and Vadhan, 2016]. The authors also demonstrate how to privately query the utility of observed parameters on private data by coupling their Laplace-based mechanism with $\mathrm{AboveThreshold}$, a classical differentially private algorithm [Dwork and Roth, 2014, Lyu et al., 2017].

While the above mechanism provides significant privacy loss savings over a baseline method that doubles the privacy loss each round, Laplace noise is unfit for many settings in which $\ell_2$-sensitivity is used for calibrating noise. Since converting from $\ell_2$-sensitivity to $\ell_1$-sensitivity[1] incurs a dimension-dependent cost, it is important to develop a noise reduction technique with Gaussian noise.

**Contributions and paper outline.** We introduce the *Brownian mechanism*, a novel approach for privately releasing a parameter vector subject to accuracy constraints. The Brownian mechanism adds correlated Gaussian noise to a risk-minimizing parameter through a Brownian motion. Noise is then iteratively stripped by moving adaptively backwards along the random walk until a suitable stopping condition is met, such as meeting a target accuracy on a public dataset. In Section 3, we define the Brownian mechanism and characterize its privacy loss. Using machinery from martingale theory, we construct *privacy boundaries* for the Brownian mechanism — upper bounds on privacy loss that hold simultaneously with high probability. In particular, the failure probability of these bounds does not depend on the number of outcomes observed, overcoming a seeming need for a union bound faced by Ligett et al. [2017]. These privacy boundaries yield provable, high-probability bounds on privacy loss under data-dependent stopping conditions.

If private data is used to evaluate risk, then the data-dependent stopping conditions can themselves leak information. To counter this, we introduce $\mathrm{ReducedAboveThreshold}$ in Section 5, a generalization of the classical $\mathrm{AboveThreshold}$ algorithm for privately querying accuracy on sensitive data. We show how to couple $\mathrm{ReducedAboveThreshold}$ and the Brownian mechanism so that a data analyst only ever incurs *twice* the privacy loss they would incur if they had queried accuracy on a public dataset. This is in contrast to the results in Ligett et al. [2017], which note that the privacy loss of $\mathrm{AboveThreshold}$ often dominates the privacy loss incurred from using noise reduction.

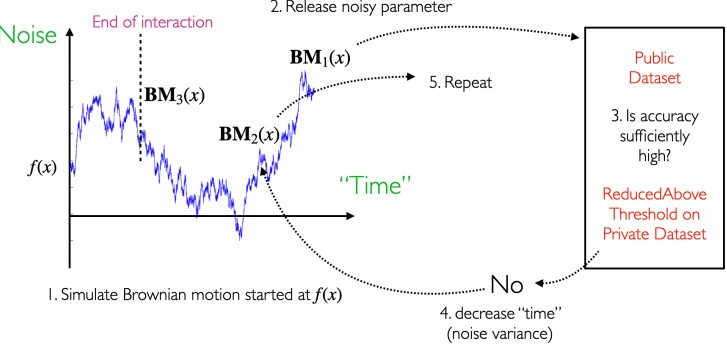

Figure 1: An example of running the Brownian mechanism to gradually release a statistic $f(x)$. First, a very noisy version of the hidden parameter $\mathrm{BM}_1(x)$ is viewed. Then, loss is measured, either on a public dataset, or on a private dataset using a method such as $\mathrm{ReducedAboveThreshold}$. If a target loss is met, the process stops. Otherwise, noise is removed and the process repeats.

---

[1]The $\ell_p$ sensitivity of $f$ is defined as $\sup_{x \sim x'} ||f(x) - f(x')||_p$ for $p \geq 1$.

We empirically evaluate the Brownian mechanism and $\mathrm{ReducedAboveThreshold}$ in Section 6, finding that the Brownian mechanism can offer privacy loss savings over the Laplace noise reduction method introduced by Ligett et al. [2017]. In our view, these results demonstrate that the Brownian mechanism is a practical, intuitive mechanism for meeting accuracy requirements in private ERM.

Lastly, we derive other new mechanisms for noise reduction, of independent interest. We generalize the Laplace process of Koufogiannis et al. [2017] to continuous time in Section 4, thus making the Laplace noise reduction mechanism of Ligett et al. [2017] more flexible and adaptive to data-dependent privacy levels. We also briefly mention a noise reduction mechanism for Skellam noise in Appendix **??**, a discrete distribution used in count queries [Agarwal et al., 2021].

## 2    Preliminaries

**Differential privacy, privacy loss, and ex-post privacy.** An algorithm $A : \mathcal{X} \to \mathcal{Y}$ is $(\epsilon, \delta)$-differentially private if, for any measurable set $E \subset \mathcal{Y}$ and any neighboring inputs $x \sim x'$,

$$\mathbb{P}(A(x) \in E) \leq e^{\epsilon} \mathbb{P}(A(x') \in E) + \delta. \tag{1}$$

In the above [Dwork et al., 2006], $\sim$ denotes some arbitrary neighboring relation. Typically $x \sim x'$ indicates $x$ and $x'$ differ in one entry, but any other relation suffices. While differential privacy has proven itself a mainstay of private computation, condition (1) is too rigid to allow data analysts to achieve a minimum desired accuracy. In other words, it embraces a *privacy first* perspective, fixing a strict condition in terms of parameters $\epsilon$ and $\delta$ that must be met. We are interested in the *accuracy first* perspective, setting a target accuracy and correspondingly optimizing privacy parameters.

The above definition of differential privacy is qualitatively focused on bounding the information-theoretic quantity of *privacy loss* [Dwork et al., 2006, 2010, Dwork and Roth, 2014].

**Definition 2.1** (**Privacy Loss**). *Let $A : \mathcal{X} \to \mathcal{Y}$ be an algorithm, and fix neighbors $x \sim x'$ in $\mathcal{X}$. Let $p^x$ and $p^{x'}$ be the respective densities of $A(x)$ and $A(x')$ on the space $\mathcal{Y}$ with respect to some reference measure[2]. Then, the privacy loss between $A(x)$ and $A(x')$ is the random variable*

$$\mathcal{L}(x, x') := \log \left( \frac{p^x(A(x))}{p^{x'}(A(x))} \right).$$

We think of $A(x)$ as the true outcome, and $\mathcal{L}(x, x')$ measures how much more likely this outcome is under the true input $x$ versus an alternative $x'$. Privacy loss provides a *probabilistic* definition of privacy. Namely, $A$ is $(\epsilon, \delta)$-*probabilistically differentially private* if, for all neighbors $x \sim x'$,

$$\mathbb{P}\left(\mathcal{L}(x, x') > \epsilon\right) \leq \delta. \tag{2}$$

While probabilistic differential privacy is not equivalent to differential privacy [Kasiviswanathan and Smith, 2014], $(\epsilon, \delta)$-probabilistically differential privacy implies $(\epsilon, \delta)$-differential privacy. Probabilistic differential privacy emerged as a means for studying privacy composition, and has been leveraged in proving many results [Kairouz et al., 2015, Murtagh and Vadhan, 2016, Rogers et al., 2016, Whitehouse et al., 2022]. A natural extension of privacy to the accuracy-oriented regime is *ex-post* privacy, which allows the bound in condition (2) to depend the observed algorithm output.

**Definition 2.2** (Ligett et al. [2017]). *Let $A : \mathcal{X} \to \mathcal{Y}$ be an algorithm and $\mathcal{E} : \mathcal{Y} \to \mathbb{R}_{\geq 0}$ a function. We say $A$ is $(\mathcal{E}, \delta)$-ex-post private if, for any neighboring inputs $x \sim x'$, we have*

$$\mathbb{P}\left(\mathcal{L}(x, x') > \mathcal{E}(A(x))\right) \leq \delta.$$

While any algorithm is trivially ex-post private with $\mathcal{E}(A(x)) := \infty$, the goal is to make $\mathcal{E}(A(x))$ as small as possible. We describe theoretical tools for obtaining ex-post privacy guarantees in Section 3, and empirically compute the ex-post privacy distributions of various mechanisms in Section 6.

**Background on Noise Reduction.** Heuristically, a noise reduction mechanism allows a data analyst to view multiple, increasingly accurate estimates of a risk minimizing parameter while only paying

---

[2]For instance, if $\mu_x$ and $\mu_{x'}$ are the laws of $A(x)$ and $A(x')$ respectively, the reference measure can be taken to be $\mu_x + \mu_{x'}$.

an ex-post privacy cost for the *least* noisy iterate observed. More formally, let $M : \mathcal{X} \to \mathcal{Y}^\infty$ be any algorithm mapping databases to sequences of outputs. The indices of the sequence denote the rounds of interaction, with smaller indices indicating greater noise. Let $M_n : \mathcal{X} \to \mathcal{Y}$ give the $n$th element of the sequence and $M_{1:n} : \mathcal{X} \to \mathcal{Y}^n$ the first $n$ elements.

**Definition 2.3** (**Noise Reduction Mechanism**). *We say $M : \mathcal{X} \to \mathcal{Y}^\infty$ is a noise reduction mechanism if, for any $n \geq 1$ and any neighboring datasets $x \sim x'$, we have*

$$\mathcal{L}_{1:n}(x, x') = \mathcal{L}_n(x, x'),$$

*where $\mathcal{L}_{1:n}(x, x')$ denotes the privacy loss between $M_{1:n}(x)$ and $M_{1:n}(x')$ and $\mathcal{L}_n(x, x')$ is the privacy loss between $M_n(x)$ and $M_n(x')$.*

The only noise reduction mechanism in the literature uses a Markov process with Laplace marginals [Koufogiannis et al., 2017] to gradually release a sensitive parameter [Ligett et al., 2017]. As originally presented, this *Laplace Noise Reduction* mechanism is nonadaptive, requiring a data analyst to fix a finite sequence of privacy parameters $(\epsilon_n)_{n \in [K]}$ in advance. Instead of presenting this method as background, we describe it in Section 4, in which we construct an adaptive generalization of this mechanism. We then leverage this generalization as a subroutine in $\mathrm{ReducedAboveThreshold}$, a generalization of $\mathrm{AboveThreshold}$ with adaptive privacy guarantees.

**Background on Brownian Motion.** We now provide a brief background on Brownian motion, perhaps the best-known example of a continuous time stochastic process [Le Gall, 2016].

**Definition 2.4.** *A continuous time real-valued process $(B_t)_{t \geq 0}$ is called a standard Brownian motion if (1) $B_0 = 0$, (2) $(B_t)_{t \geq 0}$ has continuous sample paths, (3) $(B_t)$ has independent increments, i.e. $B_{t+s} - B_s$ is independent of $B_s$ for all $s, t \geq 0$, and (4) $B_t \sim \mathcal{N}(0, t)$ for all $t \geq 0$.*

We say a process $(B_t)_{t \geq 0}$ is a $d$-dimensional standard Brownian motion if each coordinate process is an independent standard Brownian motion.

We use many properties of Brownian motion to construct the Brownian mechanism and analyze its privacy loss in Section 3. One important property of Brownian motion is that it is a continuous time martingale. This property allow us to use time-uniform supermartingale concentration to characterize and bound the privacy loss of the Brownian mechanism at data-dependent stopping times [Howard et al., 2020, 2021]. We do not go into detail about martingale concentration in this background section, but rather defer it to Appendix A. Additionally, $(B_t)_{t \geq 0}$ is a Markov process. This tells us that if we inspect the Brownian motion at times $0 \leq t_1 < t_2 < \cdots < t_n$, then $B_{t_2}, \ldots, B_{t_n}$ can be viewed as a randomized post-processing of $B_{t_1}$ that *does not* depend on $B_s$ for any $s < t_1$. This property allows us to show that the privacy loss of the Brownian mechanism — which adds noise to a parameter via a Brownian motion — only depends on the least noisy parameter observed.

## 3 The Brownian Mechanism: a Gaussian Noise Reduction Mechanism

The Brownian mechanism works by simulating a Brownian motion starting at some multivariate parameter; this parameter should be thought of as the risk-minimizing output if there were no privacy constraints. The data analyst first observes the random walk at some large time. Then, if so desired, the analyst "rewinds" time to an earlier point on the Brownian path, reducing noise to obtain a more accurate estimate. Due to the Markovian nature of Brownian motion, the analyst will only pay a privacy cost proportional to variance of the random walk at the earliest inspected time. In what follows, when we refer to a sequence $(T_n)_{n \geq 1}$ of *time functions*, we mean a sequence of functions $T_n : (\mathbb{R}^d)^{n-1} \to \mathbb{R}_{\geq 0}$ such that, for all $n$ and $\beta_{1:n} \in (\mathbb{R}^d)^n$,

$$T_{n+1}(\beta_{1:n}) \leq T_n(\beta_{1:n-1}). \tag{3}$$

**Definition 3.1.** *Let $f : \mathcal{X} \to \mathbb{R}^d$ be a function and $(T_n)_{n \geq 1}$ a sequence of time functions. Let $(B_t)_{t \geq 0}$ be a standard $d$-dimensional Brownian motion. The Brownian mechanism associated with $f$ and $(T_n)_{n \geq 1}$ is the algorithm $\mathrm{BM} : \mathcal{X} \to (\mathbb{R}^d)^\infty$ given by*

$$\mathrm{BM}(x) := \big( f(x) + B_{T_n(x)} \big)_{n \geq 1},$$

*where we set $T_n(x) := T_n \big( f(x) + B_{T_1(x)}, \ldots, f(x) + B_{T_{n-1}(x)} \big)$ with $T_1(x)$ being constant.*

We have chosen $T_n(x)$ as indexing notation to denote dependence on $x$, even if this is only through observed parameters. In the context of ERM, one can think of $f$ as computing a risk minimizing parameter associated with a private dataset $x \in \mathcal{X}$. The data analyst uses $T_n$ along with the first $n-1$ iterates to determine how far to rewind time to obtain the $n$th iterate. Due to the Markovian nature of Brownian motion, we get the following lemma. We include a proof in Appendix B for completeness.

**Lemma 3.2.** *Let $x \sim x'$ be neighbors. For any $n \geq 1$, let $\mathcal{L}_{1:n}^{\mathrm{BM}}(x, x')$ denote the privacy loss between $\mathrm{BM}_{1:n}(x)$ and $\mathrm{BM}_{1:n}(x')$ and $\mathcal{L}_n^{\mathrm{BM}}(x, x')$ the privacy loss between $\mathrm{BM}_n(x)$ and $\mathrm{BM}_n(x')$. Then,*

$$\mathcal{L}_{1:n}^{\mathrm{BM}}(x, x') = \mathcal{L}_n^{\mathrm{BM}}(x, x').$$

Lemma 3.2 just tells us that the Brownian mechanism is a noise reduction mechanism, i.e. that the privacy lost by viewing the first $n$ iterates is exactly the privacy lost by viewing the $n$th iterate in isolation. Thus, we can identify $\mathcal{L}_{1:n}^{\mathrm{BM}}(x, x')$ with $\mathcal{L}_n^{\mathrm{BM}}(x, x')$ going forward.

The Brownian mechanism, as defined above, produces an infinite sequence of parameters. In practice, a data analyst will only view finitely many iterates, stopping when some utility condition has been met or a minimum privacy level is reached. We introduce *stopping functions* to model how a data analyst adaptively interacts with noise reduction mechanisms.

**Definition 3.3** (**Stopping Function**). *Let $M : \mathcal{X} \to \mathcal{Y}^\infty$ be a noise reduction mechanism. For $x \in \mathcal{X}$, let $(\mathcal{F}_n(x))_{n \in \mathbb{N}}$ be the filtration given by $\mathcal{F}_n(x) := \sigma(M_i(x) : i \leq n)$.[3] A function $N : \mathcal{Y}^\infty \to \mathbb{N}$ is called a stopping function if for any $x \in \mathcal{X}$, $N(x) := N(M(x))$ is a stopping time with respect to $(\mathcal{F}_n(x))_{n \geq 1}$.*

A stopping function $N$ is a rule used to decide when to stop viewing parameters that *only* depends on the observed iterates of the noise reduction mechanism. $N$ could heuristically be "stop at the first time a parameter achieves an accuracy of 95% on a held-out dataset." If a data analyst uses a stopping function alongside BM, per Definition 2.3, the privacy loss accrued upon stopping is $\mathcal{L}_{N(x)}^{\mathrm{BM}}(x, x')$. Recall from Figure 1 and equation (3) that the later iterations of BM correspond to smaller noise variances, meaning that $T_n$ is a decreasing sequence in the number of iterations $n$. Further, the filtration $\mathcal{F}$ defined above is quite different from the usual filtrations considered for Brownian motions. In some cases, an analyst may want the stopping function to depend on the underlying private dataset through more than just the released parameters, e.g. they may want their rule to be "stop at the first time a parameter achieves an accuracy of 95% on the private dataset." In this case, additional privacy may be lost due to observing $N(x)$. We detail how to handle this more subtle case in Section 5.

The following theorem characterizes the privacy loss of the Brownian mechanism.

**Theorem 3.4.** *Let $\mathrm{BM}$ be the Brownian mechanism associated with $(T_n)_{n \geq 1}$ and a function $f$. For neighbors $x \sim x'$, the privacy loss between $\mathrm{BM}_{1:n}(x)$ and $\mathrm{BM}_{1:n}(x')$ is given by*

$$\mathcal{L}_{1:n}^{\mathrm{BM}}(x, x') = \mathcal{L}_n^{\mathrm{BM}}(x, x') = \frac{||f(x) - f(x')||_2^2}{2T_n(x)} + \frac{||f(x) - f(x')||_2}{T_n(x)} W_{T_n(x)},$$

*where $(W_t)_{t \geq 0}$ is a standard, univariate Brownian motion. Suppose $f$ has $\ell_2$-sensitivity at most $\Delta_2$. Then, letting $a^+ := \max(0, a)$, we have*

$$\mathcal{L}_n^{\mathrm{BM}}(x, x') \leq \frac{\Delta_2^2}{2T_n(x)} + \frac{\Delta_2}{T_n(x)} W_{T_n(x)}^+.$$

Theorem 3.4 also holds when a deterministic time $n$ is replaced by $N(x)$, where $N$ is a stopping function. The above theorem can be viewed as a process-level equivalent of the well-known fact that the privacy loss of the Gaussian mechanism has an uncentered Gaussian distribution [Balle and Wang, 2018]. We prove the Theorem 3.4 in Appendix B. Given the clean characterization of privacy loss above, we now show how to construct high-probability, time-uniform privacy loss bounds. We define *privacy boundaries*, which map the variance of BM to high-probability bounds on privacy loss.

---

[3]The notation $\sigma(X)$ denotes the $\sigma$-algebra generated by $X$. $N$ is said to be a stopping time with respect to $(X_n)$ if $\{N \leq n\} \in \sigma(X_m : m \leq n)$ for all $n \in \mathbb{N}$. This definition can be extended to allow for $N$ to depend on independent, external randomization, but we omit this for simplicity.

**Definition 3.5.** *A function $\psi : \mathbb{R}_{\geq 0} \to \mathbb{R}_{\geq 0}$ is a $\delta$-privacy boundary for the Brownian mechanism associated with time functions $(T_n)_{n \geq 1}$ if for any neighboring datasets $x \sim x'$, we have*

$$\mathbb{P}\left(\exists n \geq 1 : \mathcal{L}_n^{\mathrm{BM}}(x, x') > \psi(T_n(x))\right) \leq \delta.$$

Since the privacy loss of BM is a deterministic function of a Brownian motion, we can apply results from martingale theory to construct general families of privacy boundaries.

**Theorem 3.6.** *Assume the same setup as in Theorem 3.4. Let $\delta > 0$ and $f$ be a function with $\ell_2$-sensitivity $\Delta_2$. The following classes of functions form $\delta$-privacy boundaries.*

1. *(**Mixture boundary**) For any $\rho > 0$, $\psi_\rho^M$ given by*

$$\psi_\rho^M(t) := \frac{\Delta_2^2}{2t} + \frac{\Delta_2}{t}\sqrt{2(t + \rho)\log\left(\frac{1}{\delta}\sqrt{\frac{t + \rho}{\rho}}\right)}.$$

2. *(**Linear boundary**) For any $a, b > 0$ such that $2ab = \log(1/\delta)$, $\psi_{a,b}^L$ given by*

$$\psi_{a,b}^L(t) := \frac{\Delta_2}{t}\left(\frac{\Delta_2}{2} + b\right) + \Delta_2 a.$$

We prove Theorem 3.6 in Appendix B. In the same appendix, we plot the boundaries in Figure 4.

Privacy boundaries serve a dual purpose for the Brownian mechanism. First, since time-uniform concentration bounds are valid at arbitrary data-dependent times, that need not be stopping times with respect to the standard forward Brownian Motion filtration [Howard et al., 2021], privacy boundaries provide ex-post privacy guarantees. Second, in many settings, it may be more natural for a data analyst to adaptively specify target privacy levels instead of noise levels. This is, for instance, the case in our experiments in Section 6. By inverting privacy boundaries, data analysts can compute the proper amount of noise to remove at each step to meet target privacy levels.

We make the above precise in Corollary 3.7. In what follows, when we refer to a sequence $(\mathcal{E}_n)_{n \geq 1}$ of *privacy functions*, we mean a sequence of functions $\mathcal{E}_n : (\mathbb{R}^d)^{n-1} \to \mathbb{R}_{\geq 0}$ such that, for all $n$ and $\beta_{1:n} \in (\mathbb{R}^d)^n$, $\mathcal{E}_{n+1}(\beta_{1:n}) \geq \mathcal{E}_n(\beta_{1:n-1})$.

**Corollary 3.7.** *Let $N$ be a stopping function, as in Definition 3.3. If $\psi$ is a $\delta$-privacy boundary for BM, we have*

$$\sup_{x \sim x'} \mathbb{P}\left(\mathcal{L}_{N(x)}^{\mathrm{BM}}(x, x') \geq \psi\left(T_{N(x)}(x)\right)\right) \leq \delta,$$

*i.e. the algorithm $\mathrm{BM}_{1:N(\cdot)}(\cdot)$ is $\left(\psi(T_{N(\cdot)}(\cdot)), \delta\right)$-ex post private, where $(\cdot)$ denotes a positional argument for an input $x \in \mathcal{X}$. Further, let $(\mathcal{E}_n)_{n \geq 1}$ be a sequence of privacy functions, and define*

$$T_n(\beta_{1:n-1}) := \inf\left\{t \geq 0 : \psi(t) \geq \mathcal{E}_n(\beta_{1:n-1})\right\}.$$

*Then $\mathrm{BM}_{1:N(\cdot)}(\cdot)$ is $(\mathcal{E}_{N(\cdot)}(\cdot), \delta)$-ex post private, where $\mathcal{E}_n(x)$ is defined analogously to $T_n(x)$.*

Again, $N$ should be thought of as a stopping rule based on parameter accuracy. $\mathcal{E}_n$ should be thought of as a rule for choosing the $n$th privacy parameter given $\mathrm{BM}_{1:n-1}(x)$.

## 4 An Adaptive, Continuous-Time Extension of Laplace Noise Reduction

Here, we generalize the original noise reduction mechanism of Ligett et al. [2017], which will be used as a subroutine in Algorithm 1 in the following section. We first describe the original Laplace-based Markov process of Koufogiannis et al. [2017]. Fix any positive integer $K$ and any finite, increasing sequence of times $(t_n)_{n \in [K]}$. Let $(\zeta_n)_{n=0}^K$ be the $d$-dimensional process given by $\zeta_0 = 0$ and

$$\zeta_n = \begin{cases} \zeta_{n-1} & \text{with probability } \left(\frac{t_{n-1}}{t_n}\right)^2 \\ \zeta_{n-1} + \mathrm{Lap}(t_n) & \text{otherwise.} \end{cases} \tag{4}$$

Koufogiannis et al. [2017] show that $\zeta_n \sim \mathrm{Lap}(t_n)$ and that $(\zeta_n)_{n=0}^K$ is Markovian. Ligett et al. [2017] use the above process to construct a noise reduction mechanism. Namely, they define the

the *Laplace Noise Reduction* mechanism associated with $f : \mathcal{X} \to \mathbb{R}^d$ and $(t_n)_{n \in [K]}$ to be the algorithm $\mathrm{LNR} : \mathcal{X} \to (\mathbb{R}^d)^K$ given by $\mathrm{LNR}(x) := (f(x) + \zeta_K, \ldots, f(x) + \zeta_1)$. If $t_n := \Delta_1/\epsilon_n$, then releasing $n$th component $\mathrm{LNR}_n(x)$ in isolation is equivalent to running the classical Laplace mechanism with privacy level $\epsilon_n$.

We now extend the process $(\zeta_n)_{n \in [K]}$ to a continuous time process with the same finite-dimensional distributions. Let $\eta > 0$ be arbitrary, and let $(P_t)_{t \geq \eta}$ be an inhomogeneous Poisson process with intensity function $\lambda(t) := \frac{2}{t}$. For $n \geq 1$, let $\mathcal{T}_n := \inf\{t \geq \eta : P_t \geq n\}$ be the $n$th jump of $(P_t)_{t \geq \eta}$ and set $\mathcal{T}_0 := \eta$. Noting that $P_t$ must be a nonnegative integer, define the process $(Z_t)_{t \geq \eta}$ by

$$Z_t := \sum_{n=0}^{P_t} \mathrm{Lap}(\mathcal{T}_n). \tag{5}$$

It is immediate that $(Z_t)_{t \geq \eta}$ is Markovian. We show in Appendix D that $Z_t \sim \mathrm{Lap}(t)$. With $(Z_t)_{t \geq \eta}$, one can make LNR fully adaptive, meaning that the times $(t_n)_{n \in [K]}$ at which it is invoked need not be prespecified, and can depend on the underlying input database $x$ by using time functions.

**Definition 4.1.** *Let $f : \mathcal{X} \to \mathbb{R}^d$ be a function and $(T_n)_{n \geq 1}$ a sequence of time functions. Let $(Z_t)_{t \geq \eta}$ be the process defined in Equation (5). The Laplace noise reduction mechanism associated with $f$ and $(T_n)_{n \geq 1}$ is the algorithm $\mathrm{LNR} : \mathcal{X} \to (\mathbb{R}^d)^\infty$ given by*

$$\mathrm{LNR}(x) := \big(f(x) + Z_{T_n(x)}\big)_{n \geq 1},$$

*where again $T_n(x) := T_n(f(x) + Z_{T_1(x)}, \ldots, f(x) + Z_{T_{n-1}(x)})$.*

If the analyst would prefer instead to specify privacy functions $(\mathcal{E}_n)_{n \geq 1}$, they can do so by leveraging the corresponding time functions $T_n(x) := \Delta_1/\mathcal{E}_n(x)$, where $\mathcal{E}_n(x)$ is defined analogously to $T_n(x)$. We leverage LNR in our experiments in Section 6 and the process $(Z_t)_{t \geq 0}$ as a subroutine in constructing $\mathrm{ReducedAboveThreshold}$. LNR admits the following trivial ex-post privacy guarantee.

**Proposition 4.2.** *Let $\mathrm{LNR}$ be associated with $(T_n)_{n \geq 1}$ and a function $f$ with $\ell_1$-sensitivity $\Delta_1$. If $N$ is stopping function, the algorithm $\mathrm{LNR}_{1:N(\cdot)}(\cdot)$ is $(\Delta_1/T_{N(\cdot)}(\cdot), 0)$-ex post private.*

**Skellam Noise Reduction.** Last, we briefly discuss how to generate a noise reduction mechanism for Skellam noise [Agarwal et al., 2021]. Recall that a random variable $X$ has a Skellam distribution with parameters $\lambda_1$ and $\lambda_2$ if $X =_d Y_1 - Y_2$, where $Y_1 \sim \mathrm{Poisson}(\lambda_1)$ and $Y_2 \sim \mathrm{Poisson}(\lambda_2)$ are independent Laplace random variables. For succinctness, we write $X \sim \mathrm{Skell}(\lambda_1, \lambda_2)$.

Let $(P_1(t))_{t \geq 0}$ and $(P_2(t))_{t \geq 0}$ be two independent, homogeneous Poisson process with rates $\lambda_1$ and $\lambda_2$ respectively. Observe that the continuous time process $(X_t)_{t \geq 0}$ given by $X_t := P_1(t) - P_2(t)$ is clearly Markovian, has independent increments, and has $X_t \sim \mathrm{Skell}(t\lambda_1, t\lambda_2)$. Thus, $(X_t)_{t \geq 0}$ can be used to define a Skellam noise reduction mechanism by releasing $(f(x) + X_{T_n(x)})_{n \geq 1}$ for some sequence of time functions $(T_n)_{n \geq 1}$.

## 5 Privately Checking if Accuracy is Above a Threshold

In Section 3 we presented the Brownian mechanism, characterized its privacy loss, and showed how to obtain ex-post privacy guarantees for arbitrary stopping functions. In particular, these stopping functions could be based on the accuracy of the observed iterates on public held-out data.

However, one may desire to privately check the accuracy of observed iterates on the dataset $x \in \mathcal{X}$. Ligett et al. [2017] were able to accomplish this goal by coupling LNR with $\mathrm{AboveThreshold}$, a classical algorithm for privately answering threshold queries [Dwork and Roth, 2014]. In the context of ERM, $\mathrm{AboveThreshold}$ iteratively checks if the empirical risk of each parameter is below a target threshold, stopping at the first such occurrence. The downside to $\mathrm{AboveThreshold}$ is that it requires a prefixed privacy level. In empirical studies, Ligett et al. [2017] found this fixed privacy cost dominated the ex-post privacy guarantees, showing little benefit to using noise reduction.

Below, we construct $\mathrm{ReducedAboveThreshold}$, a generalization of $\mathrm{AboveThreshold}$ which provides ex-post privacy guarantees. We show how to couple BM with $\mathrm{ReducedAboveThreshold}$ to obtain tighter ex-post privacy guarantees than coupling with $\mathrm{AboveThreshold}$ would permit. In particular, if BM is run using parameters $(\epsilon_n)_{n \geq 1}$ and $\mathrm{ReducedAboveThreshold}$ indicates the $N$th

parameter obtains sufficiently high accuracy, the privacy loss of the net procedure will be at most $2\epsilon_N$ — only twice the privacy loss that would be accrued by testing on public data.

---

**Algorithm 1** ReducedAboveThreshold (via Laplace Noise Reduction)

---

**Require:** Algorithm $\text{Alg} : \mathcal{X} \to \mathcal{Y}^\infty$, parameter $\epsilon_{\max} > 0$, threshold $\tau$, database $x \in \mathcal{X}$, utility $u : \mathcal{Y} \times \mathcal{X} \to \mathbb{R}$ where $u(\beta, \cdot)$ is $\Delta$-sensitive $\forall \beta$, privacy functions $(\mathcal{E}_n)_{n \geq 1}$ with $\mathcal{E}_n \leq \epsilon_{\max}$ $\forall n$.
   **for** $n \geq 1$ **do**
      $\epsilon_n := \mathcal{E}_n(\text{Alg}_{1:n-1}(x))$, $T_n := 2\Delta/\epsilon_n$
      $\zeta_n := Z_{T_n}$, where $(Z_t)_{t \geq \eta}$ in Eq. (5) defines the LNR mechanism with $\eta := 2\Delta/\epsilon_{\max}$.
      $\xi_n \sim \text{Lap}\left(\frac{4\Delta}{\epsilon_n}\right)$
      **if** $u(\text{Alg}_n(x), x) + \xi_n \geq \tau + \zeta_n$ **then**
         Print 1 and HALT
      **else**
         Print 0

---

$\tau$ should be seen as a target accuracy, Alg as a mechanism for releasing a parameter (e.g. BM, LNR), and $u$ as evaluating the accuracy of $\text{Alg}_n(x)$ on $x$. $\epsilon_{\max}$ is an arbitrarily large constant, representing the minimum level of privacy required, used to prevent the user from examining $(Z_t)$ at arbitrarily small times. The above generalizes to sequences of thresholds $(\tau_n)_{n \geq 1}$ and sequences $(u_n)_{n \geq 1}$ of functions $u_n : \mathcal{Y}^n \times \mathcal{X} \to \mathbb{R}$ that are $\Delta$-sensitive in their second argument, but the added generality yields only marginal benefits. When $\mathcal{E}_n = \epsilon$ for all $n$, Algorithm 1 recovers AboveThreshold as a special case. The intuition behind ReducedAboveThreshold is that by gradually removing Laplace noise from the threshold, a data analyst can ensure that privacy of the whole procedure only depends on the magnitude of Laplace noise added when the algorithm halts. The following characterizes the privacy loss of Algorithm 1.

**Theorem 5.1.** *For any $n \geq 1$ and neighboring datasets $x \sim x'$, let $\mathcal{L}_{1:n}^{\text{Alg}}(x, x')$ denote the privacy between $\text{Alg}_{1:n}(x)$ and $\text{Alg}_{1:n}(x')$. For any $x \in \mathcal{X}$, define $N(x)$ to be the first round where* ReducedAboveThreshold *run on input $x \in \mathcal{X}$ outputs 1, that is*

$$N(x) := \inf\{n \geq 1 : \text{ReducedAboveThreshold}_n(x) = 1\}.$$

*Then, the privacy loss between* ReducedAboveThreshold$(x)$ *and* ReducedAboveThreshold$(x')$, *denoted $\mathcal{L}^{\text{RAT}}(x, x')$, is bounded by*

$$\mathcal{L}^{\text{RAT}}(x, x') \leq \mathcal{L}_{1:N(x)}^{\text{Alg}}(x, x') + \mathcal{E}_{N(x)}(\text{Alg}_{1:N(x)-1}(x)).$$

We prove Theorem 5.1 in Appendix C, where we also provide a utility guarantee for ReducedAboveThreshold. This utility guarantee, much like the utility guarantee for AboveThreshold, is in practice weak as it derives from a union bound. Using Theorem 5.1, we can simply choose $\text{Alg} = \text{BM}$ as a means of adaptively generating parameters. The following corollary, which follows immediately from the above theorem, provides the ex-post privacy guarantees of combining ReducedAboveThreshold and BM.

**Corollary 5.2.** *Let* BM *be the Brownian mechanism associated with a function $f$, decreasing time functions $(T_n)_{n \geq 1}$, and a a $\delta$-privacy boundary $\psi$. Let* ReducedAboveThreshold *be run with privacy functions $(\psi(T_n))_{n \geq 1}$, threshold $\tau$, and algorithm* BM. *Then,* ReducedAboveThreshold *is $\left(2\psi(T_{N(\cdot)}(\cdot)), \delta\right)$-ex post private.*

## 6 Experiments

**Choice of tasks:** We compare the performance of BM and LNR on the tasks of regularized logistic regression via output perturbation [Chaudhuri et al., 2011] and ridge regression via covariance perturbation [Smith et al., 2017].[4] For logistic regression, we leveraged the KDD-99 dataset [KDD, 1999] with $d = 38$ features, predicting whether network events can be classified as "normal" or "malicious". For ridge regression, we used the Twitter dataset [Kawala et al., 2013] with $d = 77$

---

[4]The two tasks use the logistic loss $\ell(y, z) := \log(1 + \exp(-yz))$ and the squared loss $\ell(y, z) := \frac{1}{2}(z - y)^2$. The regularized loss on a dataset $\mathcal{D} := \{(x_i, y_i)\}_{i \in [n]}$ is $L(\beta, \mathcal{D}) := \frac{1}{n} \sum_{i=1}^{n} \ell(y_i, \beta^T x_i) + \frac{\lambda \|\beta\|_2^2}{2}$.

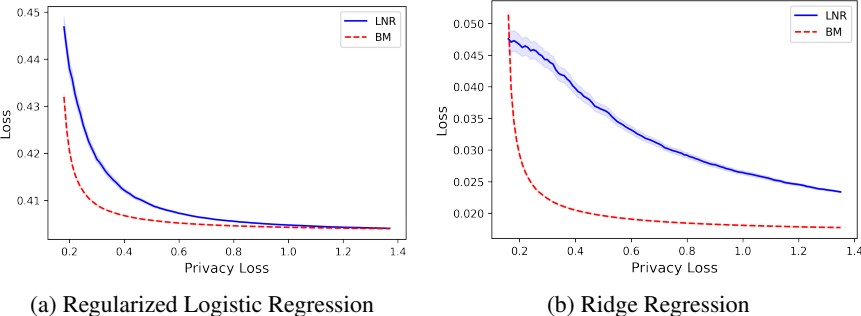

(a) Regularized Logistic Regression        (b) Ridge Regression

Figure 2: Privacy loss plotted against loss (respectively regularized logistic and ridge loss) for the statistical tasks of regularized logistic regression and ridge regression.

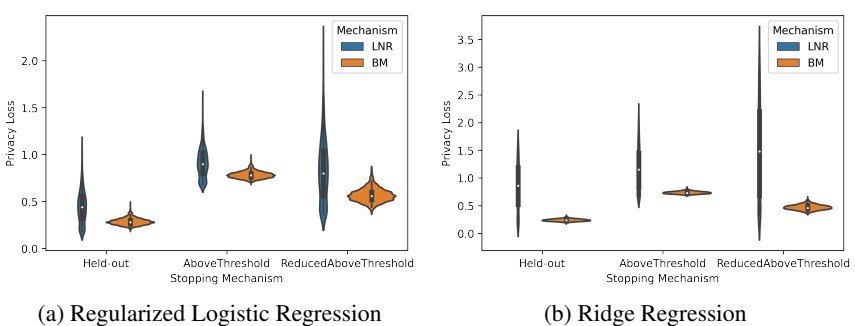

(a) Regularized Logistic Regression        (b) Ridge Regression

Figure 3: Empirical privacy loss distributions for logistic regression and ridge regression with loss assessed either (left) on the training data treated as a public, held-out dataset, (middle) via AboveThreshold, or (right) via ReducedAboveThreshold.

features to predict log-popularity of posts. In each case, we ran our experiments on $n = 10,000$ randomly sub-sampled data points. In order to guarantee bounded sensitivity, we normalized each data point to have unit $\ell_2$ norm. We note that this aspect differs from the experimentation conducted by Ligett et al. [2017], who normalized by the *maximum* $\ell_2$ norm, a non-private operation.

**Experiments:** For each task, we conducted two experiments. We discuss the specific parameter settings for these experiments in Appendix E. In the first experiment, we plotted guaranteed (in the case of LNR) or high-probability (in the case of BM) privacy loss on the x-axis against average loss (either logistic or ridge) on the y-axis. We conduct such a comparison as probability 1 privacy loss bounds cannot be provided for the Gaussian mechanism. Likewise, adding a probability $\delta$ of minimally improves privacy loss for the Laplace mechanism. We computed the average loss curve for each mechanism over 1,000 trials, and have included point-wise valid 95% confidence intervals.

In the second experiment, we plotted the empirical privacy loss distributions for BM and LNR under the stopping conditions of loss being at most 0.41 for logistic regression and 0.025 for ridge regression. For each mechanism, we evaluated this empirical distribution using three approaches for testing empirical loss: treating the training data as a held-out dataset, using AboveThreshold, and using our mechanism, ReducedAboveThreshold. In AboveThreshold, we set the privacy parameter to be fixed at $\epsilon = 0.5$. In ReducedAboveThreshold, we took the sequence of privacy parameters to be the same as the sequence of privacy parameters used by BM and LNR. We once again computed these empirical distributions over 1,000 runs of each mechanism.

**Findings:** The findings of the two experiments are summarized in Figure 2 and Figure 3. For both tasks, BM obtains significant improvements in loss over LNR near the privacy loss level that was optimized for. For both tasks, the privacy loss distribution for BM has lower median privacy loss than that of LNR. In addition, the privacy loss distribution for BM is more tightly concentrated around the median, indicating more consistent performance. The privacy loss distribution for LNR has a

heavy tail, demonstrating that many runs do not attain the target loss until high privacy loss costs are incurred. Comparing $\mathrm{ReducedAboveThreshold}$ and $\mathrm{AboveThreshold}$, we see that the privacy loss distribution for $\mathrm{ReducedAboveThreshold}$ has higher variance than that of $\mathrm{AboveThreshold}$. However, $\mathrm{ReducedAboveThreshold}$ attains a significantly lower median level of privacy loss when coupled with BM. This latter point reflects the observations of Ligett et al. [2017], who note that when $\mathrm{AboveThreshold}$ is used to determine stopping conditions on private data, it contributes the bulk of the privacy loss to the empirical distributions. On the other hand, our figures demonstrate that $\mathrm{ReducedAboveThreshold}$ results in a more mild privacy loss at target stopping conditions.

## 7 Conclusion

In this paper, we constructed the Brownian mechanism (BM), a novel approach to noise reduction that adds noise to a hidden parameter via a Brownian motion. We not only precisely characterized the privacy loss of the Brownian mechanism, but also bounded it through applying machinery from continuous time martingale theory. We then demonstrated how the utility of the iterates produced by BM can be assessed on private data via $\mathrm{ReducedAboveThreshold}$, a generalization of the classical $\mathrm{AboveThreshold}$ algorithm. This was itself accomplished by a continuous-time generalization of the original Laplace noise reduction (LNR) mechanism. Last, we empirically demonstrated that BM outperforms LNR on common statistical tasks, such as regularized logistic and ridge regression.

We comment on several limitations and open problems related to our work. We considered noise reduction mechanisms in the setting of one-shot privacy, in which only a single mechanism is run on private data. Traditional composition results, such as those for fixed privacy parameters [Dwork et al., 2010, Kairouz et al., 2015, Murtagh and Vadhan, 2016] or adaptively selected parameters [Rogers et al., 2016, Feldman and Zrnic, 2021, Whitehouse et al., 2022] are not directly applicable to algorithms satisfying ex-post privacy; additional machinery needs to be developed to handle composition in this case. A naive approach to composition is possible, which involves summing the ex-post privacy guarantees of composed algorithms and summing the corresponding $\delta$'s, but we expect this approach to be loose. Finally, noise reduction is currently only applicable to output perturbation methods; it remains open to see how to combine noise reduction with other prominent methods for private computation, such as objective perturbation.

## 8 Acknowledgements

AR acknowledges support from NSF DMS 1916320 and an ARL IoBT CRA grant. Research reported in this paper was sponsored in part by the DEVCOM Army Research Laboratory under Cooperative Agreement W911NF-17-2-0196 (ARL IoBT CRA). The views and conclusions contained in this document are those of the authors and should not be interpreted as representing the official policies, either expressed or implied, of the Army Research Laboratory or the U.S. Government. The U.S. Government is authorized to reproduce and distribute reprints for Government purposes notwithstanding any copyright notation herein.

ZSW and JW were supported in part by the NSF CNS2120667, a CyLab 2021 grant, a Google Faculty Research Award, and a Mozilla Research Grant.

JW acknowledges support from NSF GRFP grants DGE1745016 and DGE2140739.

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
