# OpenReview forum: "Brownian Noise Reduction: Maximizing Privacy Subject to Accuracy Constraints"
_NeurIPS.cc/2022/Conference — NeurIPS 2022 Accept_

### Official Review · Reviewer_4Xi3 · 2022-07-05

**Rating:** 7
**Confidence:** 4
**Soundness:** 4 excellent
**Presentation:** 4 excellent
**Contribution:** 3 good

**Summary:**

This paper generalizes and improves the framework Ligett et al. for releasing a private
query subject to accuracy constraints. In this framework, a query is released
with gradually less noise until a desired accuracy constraint is met, and the
privacy cost is the cost of the last output.

They noise the query with a continuous
distribution specified by Brownian motion, allowing the fully-adaptive release
of the query where even the desired epsilon values are allowed to depend
previous outputs. Their Brownian Motion Noise Reduction offers improved
performance over the Laplace Noise Reduction, the previous state of the art.

Finally, they generalize NoisyAboveThreshold to allow adaptive values of
epsilon. Combined with the previous results, this enables the accuracy
constraints to also depend on the the private database.


**Questions:**

Definition 3 does not make sense. What does the symbol $\sigma$ mean here? Furthermore,
what precisely is a stopping time with respect to a sequence of mechanism
outputs?

Is it efficient to produce samples from the Brownian motion algorithm or the Laplace
Noise Reduction technique? It would help the paper to describe implementation details.


**Limitations:**

Yes - there are no real limitations to this work.

**Strengths And Weaknesses:**

+ Results allow fully adaptive release of the query (where epsilon are allowed
to vary) and apply to functions with bounded l_2 sensitivity.

+ Brownian motion is simpler, more elegant, and offers improved performance over previous work.

+ The generalization of AboveThreshold allows the accuracy constraint to depend
on the private database and have the same privacy cost as releasing the final
output. This overcomes a major hurdle of previous work in this area, and makes
the accuracy-first framework more practical.

- Some minor writing mistakes, which I will describe.

---

> ### Author Response · Authors · 2022-08-02
> **Response to Reviewer 4**
>
> Response to Questions:
>
> > 1. Definition 3 does not make sense. What does the symbol $\sigma$...
>
> The notation $\sigma(X)$ denotes the sigma algebra generated by the random variable $X$. Intuitively, $\sigma(X)$ denotes the set of all possible events that can involve the random variable $X$. Likewise, if is $(X_n)$ is a sequence of random variables (or equivalently a sequence of algorithm outputs), $N$ is said to be a stopping time with respect to $(X_n)_{n \geq 1}$ if the event $(N \leq n) \in \sigma(X_m : m \leq n)$ for each $n$. Intuitively, this means whether or not one has stopped by time n can be determined just from the random variables or algorithm outputs observed up to and including time $n$. We have added a formal definition of these objects to aid the reader.
>
> > 2. Is it efficient to produce samples from the Brownian motion algorithm or...
>
> It is efficient to produce samples from both mechanisms. To produce a sample from the Brownian mechanism, one first samples a normal random variable with mean 0 and variance $T_1$ (since the first time is deterministic). Given the first $n - 1$ samples, the time $T_n$ is computed. One can then sample a point at the corresponding time $T_n$ using a Brownian bridge. To sample from the Laplace Noise Reduction technique, one first generates all of the points of jump for the inhomogeneous poisson process, which can be done using standard python packages. Then, one generates independent Laplace random variables with variances according to the sampled times from the point process. That is, if a jump occurs at time $t$, then one generates a $\mathrm{Lap}(t)$ random variable. Given these variables, one can compute the value of $Z_s$ for any $s \geq \eta$. The condition $\eta > 0$ ensures that the number of jumps of the Poisson Process is finite almost surely. We have added a short description to the appendix summarizing the above information.

---

> > ### Comment · Reviewer_4Xi3 · 2022-08-04
> > **Thanks**
> >
> > Thanks you for the clarifications

---

### Official Review · Reviewer_r5ou · 2022-07-10

**Rating:** 6
**Confidence:** 4
**Soundness:** 3 good
**Presentation:** 3 good
**Contribution:** 3 good

**Summary:**

The paper considers the Brownian mechanism, a new noise reduction DP scheme that allows for "stripping off" the noise gradually until the desired accuracy is satisfied. By leveraging the Markovian nature of the Brownian noise, the privacy guarantee of releasing a sequence of queries (with different epsilons) is the same as releasing the least private one. This improves upon using independent noise, where the privacy needs to be accounted for via composition theorem. In addition, compared to the Laplace-based method [LNRWW 2017], the Brownian mechanism gives an $\ell_2$ guarantee and hence is more suitable for FL tasks. Finally, the authors combine the Brownian mechanism with ReducedAboveThreshlod, a generalization of the AboveThreshold method in [Dwork 2014], giving an adaptive (ex-post) privacy guarantee that will be at most twice that of the non-adaptive version.

**Questions:**

1. As in [LNRWW 2017], Lemma 3.2 builds on the Markov property of the Brownian mechanism. However, as in general, the time functions $T_n$ may depend on the future of the Brownian noise, i.e., $T_n = T_n(f(x), B_{T_{n-1}},...,B_{T_1})$, i am not sure whether the Markovian property still holds. In its proof, the authors claim that by using the strong Markov property of Brownian motion, one can replace the time indices with random ones. I am not sure whether this is correct or not, as to my knowledge strong Markov property only ensures $B_{T+t_1},..., B_{T+t_n}$ being Markov for a stopping time $T$. I think the authors need to include more details in the proof, as Lemma 3.2 is the basis of the Brownian mechanism.

2. The notation of the stopping function $N(x)$ is a little confusing, as it gives a false impression that the stopping time can directly depend on the private data set $x$. However, if my understanding is correct, it could only depend on the prefix of the private release $M_{[1:n-1]}(x)$. The notation of $N(x)$ could be misleading and (incorrectly) suggest that the ReducedAbove threshold is unnecessary since one could directly compare $M_n(x)$ with $f(x)$, which would still be a valid stopping time.

==========
Post rebuttal: the authors' response to Q1 has addressed my main concern, and i have modified my score accordingly.

**Limitations:**

Yes, the social impact is properly discussed.

**Strengths And Weaknesses:**

- Strengths

The paper extends the previous noise reduction technique in [LNRWW 2017] to $\ell_2$ geometry with $(\varepsilon, \delta)$-approximate DP. Having an $\ell_2$ guarantee is always preferable for FL tasks in both theory and practice. In addition, the improved AboveThreshlod method seems to be significant and allows for much tighter privacy accounting.

- Weakness

1. In general, I think the theoretical grounding looks mostly sound. However, I do have a technical question regarding Lemma 3.2 (see the "Question" section). As Lemma 3.2 is the basis of almost all results of this paper, I'd appreciate it and will increase the score accordingly if the authors can clearly clarify it.

2. In my opinion, the presentation can be improved by providing more high-level descriptions of how this work compares with [LNRWW 2017]. In particular, in Section 4, the authors extend the LNR scheme in [LNRWW 2017] to the continuous case; however, I don't think I can fully appreciate why extending LNR to the continuous case is important and what is the advantage of doing so.

3. As the privacy of the Brownian Mechanism is ex-post and is determined by the privacy functions $\mathcal{E}_n$, it seems to be unclear how to set them in practice. It would be good if the authors could briefly discuss it.

---

> ### Author Response · Authors · 2022-08-02
> **Response to Reviewer 3**
>
> Response to Weaknesses:
>
> > 1. In my opinion, the presentation can be improved by providing more high-level descriptions...
>
> We have added justification for the importance of  adaptivity in LNR in Section 4. Extending LNR to the continuous case is beneficial as it allows a data analyst to adaptively select how much noise to remove from the perturbed optimal parameter. The original Laplace-based mechanism, as leveraged by Liggett et al., requires prefixing privacy/noise levels (choosing a sequence of constants before interacting with the data), which is very restrictive. In particular, our generalization is needed for Algorithm 1, which allows the data analyst to adaptively specify privacy levels based on observed parameter iterates.
>
> > 2. As the privacy of the Brownian Mechanism is ex-post and is determined by the privacy functions...
>
> We have included a more intuitive description of privacy boundaries before the definition. To use the Brownian mechanism, a data analyst first selects a family of privacy boundaries — either linear or mixture. If the analyst has a target approximate privacy level in mind, then they should use a linear boundary. Otherwise, a mixture boundary may be preferable as it offers greater overall tightness at the cost of tightness at any specific point in time. Then, the data analyst selects the parameters for said boundaries. This can be done heuristically, or the data analyst can select the parameters to optimize the privacy boundary for tightness at a prespecified privacy level (as is done in the experiments). This optimization can be done with a simple function call.  Then, the user can either (1) evaluate the privacy boundary at a given time function to observe the privacy loss, or (2) specify a privacy function (i.e. target privacy level), computing the corresponding time as given in Corollary 3.7.
>
> Response to Questions:
>
> > 1. As in [LNRWW 2017], Lemma 3.2 builds on the Markov property of the Brownian mechanism. However, as in general, the time functions  may depend on the future of the Brownian noise...
>
> The strong Markov property is not actually needed to prove Lemma 3.2. To address the reviewers concerns, we have provided a more direct proof that directly examines the joint density of $(B_{T_1}, \dots, B_{T_n})$ at an arbitrary starting point $\mu$. This direct proof can be found in Appendix B of the rebuttal submission, in the stead of the original proof. First, it is clear that  $B_{T_1} \sim \mathcal{N}(\mu, T_1)$, as $T_1$ is just a constant function. Then, given $B_{T_1}, \dots, B_{T_{m - 1}}$, we know using a Brownian bridge that the conditional distribution of $B_{T_m}$ is $B_{T_m} \sim \mathcal{N}\left(\mu + \frac{T_m}{T_{m - 1}}(B_{T_{m - 1}} -\mu), \frac{(T_{m - 1} - T_m)T_m}{T_{m - 1}}\right)$. In short, one computes that $$p_{1:n}^\mu\left(B_{T_n}, \dots, B_{T_1}\right) \propto \exp\left( - \frac{(B_{T_1} - \mu)^2}{2T_1}\right)\prod_{m = 2}^n \exp\left(\frac{-(B_{T_m} - \mu - \frac{T_m}{T_{m - 1}}(B_{T_{m -1}} - \mu))^2}{2(T_{m - 1} - T_m)}\cdot \frac{T_{m -1}}{T_m}\right).$$
>
> One can then check the equivalence $p_{1:n}^\mu\left(B_{T_n}, \dots, B_{T_1}\right) \propto \exp\left(\frac{-(B_{T_n} - \mu)^2}{2T_n}\right)\prod_{m = 2}^n \exp\left(\frac{-(B_{T_{m - 1}} - B_{T_m})^2}{2(T_{m - 1} - T_m)}\right)$. Given this, it is clear that, for $\mu, \mu'$, the ratio of densities is $\frac{p^\mu(B_{T_n}, \dots, B_{T_1})}{p^{\mu'}(B_{T_n}, \dots, B_{T_1})} = \frac{\exp\left(\frac{-(B_{T_n} - \mu)^2}{2T_n}\right)}{\exp\left(\frac{-(B_{T_n} - \mu')^2}{2T_n}\right)}$, as desired.
>
> > 2. The notation of the stopping function  is a little confusing, as it gives a false impression...
>
> We have chosen the notation $N(x)$ as to indicate that the stopping time $N$ does indeed depend on the underlying dataset, but have chosen not to write $N(M_{1:n}(x))$ for notational ease. In the rebuttal submission, when introducing time functions (the first instance of this indexing notation), we now explicitly establish that $T_n$ depends just on the dataset $x$ through observed iterates $M_{1:n}(x)$. We hope this will ease confusion for the reader.

---

> > ### Comment · Reviewer_r5ou · 2022-08-05
> > **Thank you for the response**
> >
> > Thank you for your detailed response, and in particular, the response to my first question has addressed my main concern. I will increase my score accordingly.

---

### Official Review · Reviewer_zDCJ · 2022-07-16

**Rating:** 6
**Confidence:** 2
**Soundness:** 3 good
**Presentation:** 2 fair
**Contribution:** 3 good

**Summary:**

The paper introduces a Brownian mechanism for private release, in which an analyst (1) generates a sequence of noise values from a Brownian process up until a certain large time point, (2) starting from this large time point, takes steps back until they find a noise value which incurs acceptable utility, and (3) releases the data with noise at the identified time point. Because of the Markov properties of the process, the privacy loss of such mechanism only depends on the final noise value output. The authors show two ways to compute the epsilon-guarantee of the resulting mechanism in terms of ex-post privacy. Next, the authors extend the existing Laplace-based Markov process mechanism to continuous time. In the final part of the theoretical sections of the paper, the authors introduce a ReducedAboveThreshold mechanism for deciding when to stop the Brownian (or another noise reduction) mechanism when the utility has to be computed on private data. In this algorithm, the utility with added Laplace noise is compared against a noisy threshold, with threshold noises coming from a Laplace-based Markov process which is synchronized with the main (e.g., Brownian) process. Finally, the authors conduct two experiments on private release of the logistic regression parameter vector, and ridge regression via covariance perturbation, finding that the Brownian mechanism on average results in lower privacy loss at the same level of utility threshold, and the privacy loss is more consistent.

**Questions:**

* How does the proposed mechanism relate to the prior work on fully adaptive composition? Can the results be re-analyzed in the framework of fully adaptive composition?
* In the experiments, are comparisons of guaranteed privacy loss and high-probability loss fair? How is the privacy loss computed in the experiments (using Theorem 3.6?).
* Is the continuous-time extension of Laplace-based Markov process mechanism necessary for Algorithm 1?
* Is there a reason that the Algorithm 1 settles when finding any noise level within the utility threshold, if it seems that it could proceed to actually do the minimization of privacy loss?

**Limitations:**

The limitations are discussed.

**Strengths And Weaknesses:**

The paper deals with a useful setting in modern data analysis, in which an analyst might want to adaptively pick the best privacy level subject to accuracy constraints. The paper proposes two tools (Brownian mechanism and ReducedAboveThreshold) which formalize an iterative interactive procedure in which the analyst tries several noise levels before settling with a satisfactory one. Unlike the previous work on Laplace-based process mechanism, the proposed Brownian mechanism is calibrated to L2 sensitivity, thus suitable to multi-dimensional vectors. I believe this is a useful contribution to the field, given that the results are correct (I have not verified the proofs.)

The main issue with the paper is clarity. The notation is obtuse with the generic time functions $T$ which have an unclear definition, stopping functions $N$, privacy boundary functions $\psi$, privacy functions $\mathcal{E}$. It took a lot of time and effort to parse. The intended meaning of some functions only become clear at a much later point after its introduction, such as $T_n$ by line 211. In general, the generality of exposition is a hindrance. Corollary 2.7, if I understand the intention correctly, would be better as two algorithms for two usage scenarios (accuracy-first and privacy-first). Section 3 seems to be disconnected from the rest of the text. It is also not fully clear if the results of the paper cannot be achieved using the tools of fully adaptive composition. In the experimental section, although the results are convincing, it is unclear if the comparison of guaranteed privacy and high-probability privacy is fair, and how exactly is the empirical privacy loss computed.

As ERM is a motivation of the work, it would be interesting to see how the output perturbation with BM compares to objective perturbation at the same accuracy levels.

---

> ### Author Response · Authors · 2022-08-02
> **Response to Reviewer 2**
>
> Response to Weaknesses:
>
> > 1. The main issue with the paper is clarity...
>
> The definitions of the aforementioned objects are intended to be fully rigorous, and hence ensure the correctness of the proofs found in the paper. After each definition, there are intuitive written explanations. We had not provided such an intuitive explanation for privacy boundaries, and thus have now introduced one into the rebuttal revision of the paper.
>
> > 2. Corollary 2.7, if I understand the intention correctly, would be...
>
> We believe the reviewer means Corollary 3.7. Both usage scenarios for corollary 3.7 are intended for the accuracy-first regime. The first situation (mentioned in the first part of the corollary) provides ex-post privacy guarantees for a fixed privacy boundary. We have included the second part to show how a data analyst can adaptively select target privacy levels while using the Brownian mechanism.
>
> > 3. Section 3 seems to be disconnected from the rest of the text.
>
> We believe the reviewer meant Section 4 as opposed to Section 3. Section 4 is intended to introduce the Laplace Noise Reduction mechanism, a strict generalization of the noise reduction algorithm presented in Ligett et al. We present this mechanism in the body of the paper as it is used as a subroutine in ReducedAboveThreshold. We have added “which will be used as a subroutine in the following section” to the first line of Section 4 to further indicate connection to the rest of the paper.
>
> > 4. It is also not fully clear if the results of the paper cannot be achieved using the tools...
>
> To apply results from fully adaptive composition to this setting, fresh Gaussian noise would need to be added in each round. This would ultimately lead to a blowup in the privacy loss proportional to the number of parameters observed. The Brownian mechanism circumvents this blowup by adding strongly correlated Gaussian noise through a Brownian motion. Due to this high correlated noise, results from fully adaptive composition cannot be applied. Results from martingale concentration needed to be applied instead. The result is that the privacy loss at the end is *independent* of the number of interactions, or the number of parameters observed, as is the main point of noise reduction mechanisms.
>
> > 5. In the experimental section, although the results are convincing, it is unclear...
>
> Unfortunately, the privacy guarantees of Laplace Noise reduction gain virtually no benefit from conversion to approx DP (since the marginals of the process are Laplace). For instance, the conversion $\epsilon’ = \log(\exp(\epsilon) \cdot (1-\delta) - \delta)$ is known from Murtagh and Vadhan 2016 (https://arxiv.org/pdf/1507.03113.pdf). If one were to set  $\delta = 10^{-6}$ and $\epsilon = 1$, this yields $\epsilon’ = 0.99999863212$, a negligible improvement. Moreover, such guarantees are not time-uniform in nature, which is necessary for the application of LNR. Likewise, since the marginals of the Brownian mechanism are Gaussian, pure privacy guarantees cannot be extracted. We thus conduct such comparisons due to mathematical limitations, not limitations in our analysis. The empirical privacy loss is computed using Theorem 3.6 and Corollary 3.7. Suppose we have chosen to use a linear privacy boundary, as is the case in our experimentation. Furthermore, suppose we have fixed some associated parameters with the boundary. Empirical privacy loss is then simply computed by plugging the time at which the mechanism stopped (i.e. corresponding variance) into the privacy boundary. Over multiple runs, this quantity is then averaged.
>
> Response to Questions:
>
> > 1. How does the proposed mechanism relate to the prior work on...
>
> Please see 4 under weaknesses.
>
> > 2. In the experiments, are comparisons of guaranteed privacy loss and high-probability loss fair...
>
> Please see 5 under weaknesses
>
> > 3. Is the continuous-time extension of Laplace-based Markov...
>
> Yes, the continuous time extension of the Laplace-based Markov process is necessary for algorithm 1. This is because Algorithm 1 allows the data analyst to adaptively specify an increasing sequence of privacy levels based on the outputs of the algorithm. The Laplace-based Markov process leveraged by Ligett et al requires privacy levels that are fixed in advance of viewing the data (i.e. a sequence of constants) — thus making it incompatible with Algorithm 1. We have added mention of the necessity in Section 4.
>
> > 4. Is there a reason that the Algorithm 1 settles when finding any noise level...
>
> Algorithm 1 attempts to find the greatest (not any) noise level — hence the smallest possible epsilon — such that the target accuracy is satisfied. In other words, it minimizes the privacy loss (maximizes privacy) subject to meeting the utility threshold. If the algorithm were to be further run after the first instance of meeting a target accuracy, additional information about the underlying dataset would be leaked.

---

> > ### Comment · Reviewer_zDCJ · 2022-08-09
> > **Response**
> >
> > Thank you for the detailed response and the clarifications, and apologies for the shifted section numbering.

---

### Official Review · Reviewer_Nc8u · 2022-07-19

**Rating:** 7
**Confidence:** 5
**Soundness:** 4 excellent
**Presentation:** 4 excellent
**Contribution:** 3 good

**Summary:**

The paper introduces the theoretical framework for a new noise reduction algorithm for probabilistic differential privacy (DP), namely, the Brownian mechanism (BM), and it also compares it empirically with the Laplace noise reduction (LNR) mechanism on empirical risk minimization (ERM) tasks. The privacy loss (random variable) for BM is characterized, a bound on it is given, and two 'privacy boundaries' are derived for it using results from martingale theory. Additionally, LNR, introduced in [Ligett et al., 2017], is extended in this paper to the continuous-time setting (and it is briefly indicated in the appendices how a Skellam noise reduction mechanism can be derived from the present work).

**Questions:**

- In Line 495, for $X_t^{\pi}$ to be well-defined, one at least should require that each $(\lambda,\omega) \mapsto Y_t^\lambda(\omega)$ is a Borel function.

- What is $T_1$ in Definition 3.1? It looks like it is a pre-specified constant $T_1$ or function $x\mapsto T_1(x)$, and it would be good to add this remark. Then, do we require $T_2(f(x)+B_{T_1(x)})\le T_1$ (or $\le T_1(x)$), in accordance with the $T_n$ specifying a sequence of time functions? Various instances of such initial parts of a trend have to be carefully specified in the paper, e.g., it is imposed that $\eta>0$ in Line 227, yet it is mentioned in Line 238 that $(Z_t)_{t\ge 0}$ (i.e., $\eta=0$) will be used (perhaps one should define $\mathrm{Lap}(0)$ as deterministically $0$).

- The term 'density' is loosely mentioned in Definition 2.1 (e.g., density with respect to which measure?). I recommend slightly modifying how the definition is spelled out.

- The paper has superb presentation, but a few typos should be fixed.
- - English typos include: "depend [on] the" in Line 101; delete 'the' in Lines 187 and 222; replace ',' with '.' in Line 235; delete 'instead' in Line 570.
- - Math typos include: need $\mathcal{L}_{1:n}^{\mathrm{Alg}}$ in the last equation after Line 546; $Z(t)$ is used in Line 559, but only defined in Line 593; should probably be $\mathrm{Lap}(\tau,\frac{2\Delta}{\epsilon_n})$ in Line 572; the two equations after Line 586 need to be fixed; 'minus' sign missing in penultimate equation after Line 612; also, the wording of Prop. 4.2 might be revised, as ex-post privacy is prefixed with a pair when first defined.

I also recommend that it be mentioned that Fig. 4 is in the appendix (when mentioned in Line 198); also, a comment by the authors on why the variances shown in Fig. 4 are large might be in order.

**Limitations:**

The authors mention briefly in the conclusion that existing composition results for DP are inapplicable for BM, and that noise reduction is concerned only with output perturbation. Although not a hinderance to the quality of the paper, it would have been nice to see some indication on how composition results can potentially be derived for BM.

The authors mention that there are "no negative societal impacts," but it can be argued that privacy mechanism can be misused.

**Strengths And Weaknesses:**

Strengths:

- Quality: The derived results and the empirical assessment are of high quality and highly non-trivial.
- Clarity: The paper is extremely well-written.

More details: The introduction of BM is a natural step after LNR, as Gaussian perturbation is expected to be more private than a Laplace perturbation if one cares about $\ell_2$-sensitivity. Nevertheless, the authors borrow results from martingale theory that make the introduction of BM a nontrivial analogue of LNR. Further, the results are derived in a highly rigorous manner. Also, the extension of LNR to the continuous setting, besides being rigorously derived in a nice way, yields a significant improvement regarding the privacy problem: querying a utility now can only cause a privacy loss comparable to that incurred by the disclosure of the private parameter. Additionally, a union bound in [Ligett et al., 2017] is disposed of, yielding tighter bounds.

=====

Weaknesses: The following are not detrimental weaknesses of the paper, as it is strong as is; rather, they are rooms for improvement.

- The performance comparison between BM and LNR is largely empirical. The main concern here is that it could be that the chosen datasets and tasks fall within a regime where BM outperforms LNR, but that this regime, as far as the present paper is concerned, is unknown. The authors mention in Line 16: "[BM] empirically outperforms existing work on common statistical tasks, ...." Relatedly, in Line 302, the authors write "... we plotted guaranteed (in the case of LNR) or high-probability (in the case of BM) privacy loss on the x-axis against average loss (either logistic or ridge) on the y-axis." In addition, the authors write in Appendix C in Line 579, before proving a utility guarantee for ReducedAboveThreshold, "... instead of plotting the utility guarantee in our experiments in Section 6, we instead plot empirically observed loss/accuracy." In view of these statements, I think the work could be stronger if more aligned comparisons between BM and LNR are presented.

- There are no details about how BM can be incorporated with composition of DP mechanisms; this issue is mentioned only as a limitation in the conclusion of the paper.

---

> ### Author Response · Authors · 2022-08-02
> **Response to Reviewer 1**
>
> Response to Weaknesses:
>
> > 1. The authors mention in Line 16: "[BM] empirically outperforms existing work on common statistical tasks, ...." Relatedly, in Line 302, the authors write "...
>
> Unfortunately, the privacy guarantees of Laplace Noise reduction gain virtually no benefit from conversion to approximate privacy (since the maringals of the process are Laplace). For instance, the conversion $\epsilon’ = \log(\exp(\epsilon) \cdot (1-\delta) - \delta)$ is known from Murtagh and Vadhan 2016 (https://arxiv.org/pdf/1507.03113.pdf). If one were to set  $\delta = 10^{-6}$ and $\epsilon = 1$, this yields $\epsilon’ = 0.99999863212$, a negligible improvement. Moreover, such guarantees are not time-uniform in nature, which is necessary for the application of LNR. Likewise, since the marginals of the Brownian mechanism are Gaussian, pure privacy guarantees cannot be extracted. We thus conduct such comparisons due to mathematical limitations, not limitations in our analysis. The utility guarantees of ReducedAboveThreshold do not depend on the underlying mechanism masking the parameter (LNR or BM). As such, plotting these utility guarantees would not illustrate a difference between LNR and BM.
>
> > 2. There are no details about how BM can be incorporated with composition of DP mechanisms...
>
> We agree that the composition of mechanisms satisfying ex-post privacy is of great importance. In this paper we consider the problem of noise reduction in a one-shot setting. We will expand upon the composition limitation, noting that one can obtain a “naive” composition guarantee by adding the ex-post bounds and delta parameters. Developing a more refined theory of composition for algorithms satisfying ex-post privacy would likely necessitate new theoretical tools, and we leave this as an area of investigation for future work.
>
> Response to Questions:
>
> > 1. In Line 495, for $X^\pi_t$ to be well-defined....
>
> I believe you are correct here that $(\lambda, \omega) \mapsto Y_t^{\lambda}$ needs to be jointly measurable in $\lambda$ and $\omega$. However, the processes we consider in this paper are all of the form $Y_t^{\lambda} := \exp(\lambda M_t - \frac{\lambda^2}{2}t)$ for a martingale $(M_t)$, so the desired joint measurability assumption is trivially satisfied.
>
> > 2. What is $T_1$ in Definition 3.1...
>
> You are correct, $T_1 = T_1(x)$ should indeed be a constant function. We have explicitly added this to the definition to ensure no further confusion. We have made a typo in line 238, and have corrected it to the proper constraint that $t \geq \eta$. Thanks for bringing this to our attention. The proof for the marginal distributions of $(Z_t)_{t \geq \eta}$ requires $\eta$ to be strictly greater than 0 (but can be arbitrarily small, thus almost equal to zero for all practical purposes). This constraint is also necessary for the purposes of simulation.
>
> > 3. The term 'density' is loosely mentioned in Definition 2.1...
>
> We have added a footnote explaining which measure the densities can be assumed to be taken with respect to. In particular, we note that one can take the density to be the Radon-Nikodym derivative of the law of A(x) (or  A(x’) respectively) with respect to the sum of the laws of A(x) and A(x’).
>
> > 4. The paper has superb presentation, but a few typos should be fixed...
>
> Thank you for bringing the typos to our attention. We have gone through the paper and corrected said typos.
>
> > 5. a comment by the authors on why the variances shown in Fig. 4 are large might be in order.
>
> We have added mention that Fig. 4 is located in the appendix. We have corrected our caption to note that the privacy level optimized for in the figure was $\epsilon = 0.3$, not $\epsilon = 0.5$. Such large variances are expected. For instance, for $\delta = 10^{-6}$, to obtain a privacy level of $\epsilon = 0.3$, a standard conversion yields $\sigma^2 = 2\cdot \frac{\log(1.25/\delta)}{\epsilon^2} = 135$. Since our bounds are time uniform in nature, they naturally are a bit looser than any point-wise optimal bound.

---

> > ### Comment · Reviewer_Nc8u · 2022-08-09
> > **Thank you!**
> >
> > I thank the authors for the clarifications and edits. I encourage you to give the paper one more round of editing to correct minor typos, e.g., $(Z_t)_{t \ge 0}$ in Line 236, and note that the probabilities sum to 1 in the proof of Proposition C.1 so the total sum is $\gamma/2$. Also, it could be helpful to increase the font sizes in the plots.

---

### Meta-Review · Area_Chair_TCaY · 2022-08-25

**Recommendation:** Accept
**Confidence:** Certain

**Metareview:**

The reviewers unanimously agreed that the paper is well-motivated and the theoretical results surrounding the proposed Brownian mechanism are interesting. Initial concerns regarding presentation and clarity were assuaged after the authors' responses to the reviews. Overall, the paper is a non-trivial and valuable extension of [Ligget et al., 2017] and should be presented at the conference.

**Award:**

No

---

### Decision · Program_Chairs · 2022-09-14

Accept